# LAGSwin: Local attention guided Swin-transformer for thermal infrared sports object detection

**Hengran Meng[1], Shuqi Si[1], Bingfei Mao[1]\*, Jia Zhao[1], Liping Wu[2]**

**1** Department of Basic Course Teaching, Shandong Agriculture And Engineering University, Licheng District, Jinan, Shandong, China, **2** School of Information Engineering, Shandong Agriculture And Engineering University, Licheng District, Jinan, Shandong, China

\* maobingfei1986424@163.com

## Abstract

Compared with visible light images, thermal infrared images have poor resolution, low contrast, signal-to-noise ratio, blurred visual effects, and less information. Thermal infrared sports target detection methods relying on traditional convolutional networks capture the rich semantics in high-level features but blur the spatial details. The differences in physical information content and spatial distribution of high and low features are ignored, resulting in a mismatch between the region of interest and the target. To address these issues, we propose a local attention-guided Swin-transformer thermal infrared sports object detection method (LAGSwin) to encode sports objects' spatial transformation and orientation information. On the one hand, Swin-transformer guided by local attention is adopted to enrich the semantic knowledge of low-level features by embedding local focus from high-level features and generating high-quality anchors while increasing the embedding of contextual information. On the other hand, an active rotation filter is employed to encode orientation information, resulting in orientation-sensitive and invariant features to reduce the inconsistency between classification and localization regression. A bidirectional criss-cross fusion strategy is adopted in the feature fusion stage to enable better interaction and embedding features of different resolutions. At last, the evaluation and verification of multiple open-source sports target datasets prove that the proposed LAGSwin detection framework has good robustness and generalization ability.

## Introduction

Target detection in thermal infrared images aims to identify the position and model of objects of interest (such as pedestrians, vehicles, and indoor sports players), etc. [1]. With the successful application of deep learning technology in many fields, thermal infrared targets are based on deep learning frameworks [2]. Detection algorithms have also made significant progress in recent years. Most of the existing thermal infrared detection algorithms focus on extracting high-level feature information of target objects in thermal infrared images, ignoring the

**Data Availability Statement:** TTsports from https://www.kaggle.com/datasets/aalborguniversity/thermal-soccer-dataset. RGBsports datasets from https://www.kaggle.com/datasets/mlwhiz/detection-footballvscricketball.

FLIRs from https://www.flir.com/oem/adas/adas-dataset-form.

**Funding:** The authors received no specific funding for this work.

**Competing interests:** The authors have declared that no competing interests exist.

physical appearance properties and low-level semantics of targets. Compared with remote sensing and natural images, thermal infrared images have significant semantic ambiguity between different categories due to low resolution, and contrast [3], which significantly challenges thermal infrared target detection. Obtaining better semantic details from ratio and contrast has become a research hotspot.

To obtain better thermal infrared target detection performance and detection efficiency, the existing advanced thermal infrared target detectors are mainly two-stage RCNN framework and lightweight single-stage Yolo series [4], in which the two-stage RCNN [5, 6] consists of a region proposal network (RPN) [7] Generate high-quality regions of interest from horizontal anchors for efficient features, and utilize bounding regression boxes for regression and classification. It is worth noting that the horizontal anchor point quickly leads to a severe imbalance between the bounding box and the directional target object. At the same time, there are significant differences in scale, shape, and color between the sports target objects and when the target moves to a specific period. There may be overlapping and dense phenomena. To alleviate these problems, a Region of Interest (RoI) Rotator was recently proposed to convert horizontal anchors to rotational anchors, avoiding redundant- computations brought by many anchors. However, this ROI rotation operation is mainly used for target detection in remote sensing images and less for thermal infrared target detection tasks. The single-stage detection method is mainly based on fast efficiency. Compared with the two-stage RCNN framework, the accuracy needs to be improved.

The main contributions of the LAGSwin detection framework proposed in this paper are as follows:

- A local attention-guided Swin-transformer is designed to form mutual embedding between high-level features and low-level features; when the high-level features have insufficient representation ability, many low-level features are embedded in the high-level features. Semantics, when low-level semantic information is weak, embedding a large number of high-level semantics helps resolve the semantic ambiguity between different classes.

- A criss-cross fusion strategy is designed to make the target semantics in the low-resolution feature map have a strong representation through cross-fusion. And describe the thermal infrared target from three levels: low, medium, and high, and establish effective spatial relationships and long-term dependency. At the same time, the interaction between different hierarchical features is realized, and the spatial details of sports targets in thermal infrared images are better obtained.

- Introducing its convolutional filter in the detection stage, encoding the orientation information while reducing the inconsistency between classification and localization regression and enabling us to generate high-quality anchor and alignment features for Accurate thermal detection of sports objects in infrared images. Finally, the evaluation and demonstration are carried out on the open-source thermal infrared sports dataset and other RGB sports datasets, and the proposed LAGSwin detection framework achieves the best performance in both speed and accuracy. We design a weighted loss function to tune and optimize the proposed framework.

The rest of this article is organized as follows. The second section details the related work related to thermal infrared target detection. The next section introduces the proposed LAGSwin detection framework and describes the functions and principles of different components in detail. The presents the experimental results and analysis of different detection methods. The conclusions and next research plans are described in the last section.

# Related work

We will describe the progress and current status of thermal infrared object detection from two perspectives, one-stage, and two-stage detection methods.

## Two-stage detection methods

With the successful application of deep learning technology in many fields, object detection has developed significantly in recent years. At present, the standard thermal infrared target detection algorithms are mainly divided into two types, namely, two-stage detection and single-stage detection methods. Among them, the two-stage detection method realizes feature extraction by generating a sparse ROI set and performs boundary regression and object classification in the second stage. For example, Li et al. [8] designed a faster light-sensing two-stage RCNN detection model for the differences between optical and thermal infrared images. They discussed the feature extraction capabilities of various convolutional networks in depth. Aiming at the problem of reliable and efficient object detection in thermal infrared images, Dai X et al. [5] propose a novel object detection method based on convolutional networks, which can be optimized and predicted in an end-to-end manner. Dai et al. [9] proposed a multi-task Faster RCNN detector to evaluate the driving distance to improve driving safety. They improved the performance of thermal infrared object detection tasks by adjusting the feature extractor. Song et al. [4] created a segmentation template for the heat-generating part for the heat-generating components in the thermal-sensing image of a thermal infrared camera. They proposed a mask-based RCNN-based infrared image detection algorithm. Although the detection progress of these two-stage thermal infrared target detection algorithms is good, the model efficiency needs to be improved. At the same time, these methods often use a relatively simple convolution structure in the feature extraction stage and pay too much attention to thermal infrared images in the feature capture stage. The high-level semantics of the target ignores rich low-level semantic information.

## One-stage detection methods

Compared with the two-stage target detector, the single-stage detector acts directly on the target and does not need to generate the ROI generation stage, so it lags behind the two-stage sensor in performance. For single-stage detection algorithms, Jiang C et al. [3] used yolo to capture the feature information of targets from thermal infrared images and proposed a UAV TIR target detection framework for thermal infrared photos and videos. Considering the poor performance of RGB images at night and in complex weather conditions, Krišto M et al. [6]. used YOLOv3, a detection model suitable for RGB images, to detect targets in thermal infrared images. Its detection speed and accuracy achieved good results. Competitiveness. Based on YOLOv5, Li S et al. [10] proposed a region-free object detector YOLO-FI based on the characteristics of thermal infrared images, that is, in the feature extraction stage, the cross-stage local connection in the shallow layer (cross-stage-partial -connections, CSP) module to expand and iterate, maximizing external features to improve the representation power of these features. Li L et al. [11] considering that most ship detection algorithms use artificial features to segment visible light image blocks accurately and are limited by factors such as illumination, clouds, and atmospheric waves in practical applications, they designed a complete yolo-based complex TIRSIs Ship Detection Method (CYSDM) in the context. For thermal infrared images, Hou Z et al. [12] designed a thermal infrared target detector M-YOLO that helps to integrate global context information. In the feature extraction stage, a top-down and bottom-up parallel feature fusion method is used, and the maximum limit is preserved. The representation ability of the feature is enhanced. To make up for the inability of traditional cameras to be used under

harsh lighting conditions, Li W et al. [13] designed a nighttime thermal infrared pedestrian detection algorithm through Yolov3. Xue Y et al. [14] used compressed Darknet53 to obtain the feature information of two modalities. They used a weighted fusion strategy for feature fusion, proposing a thermal infrared pedestrian detection algorithm with multi-modal attention fusion. Although these thermal infrared detection methods have good detection efficiency, the detection accuracy needs to be improved. At the same time, they mainly use traditional convolution methods or simple weighted fusion strategies in the feature extraction stage, which are often quickly introduced in the feature transfer process—a large amount of redundant information. In addition, these methods mainly focus on detecting single structural targets in thermal infrared images and less on multi-category sports targets.

Sports objects may overlap, occlude or shadow during motion. Therefore, capturing more physical appearance attribute information, such as shape and size, in the feature extraction process is necessary to enhance high-level semantics. For example, Masuda T et al. [15] proposed a motion video behavior detection method based on self-supervised feature learning and target detection, which introduced target detection into the process and realized the action detection of multiple people by tracking each person. Considering the high coupling between different features, Zhao J et al. [16] designed a non-global attention mechanism: a local u-shaped attention decoupling network. Jiang X et al. [17] propose a new complementary transformer network (MCNet) for object detection in RGB and thermal infrared images, that is, introduce a transformer-based feature extraction module to efficiently extract hierarchical features of RGB and thermal images and attention-based feature interaction and serial multi-scale dilated convolution (SDC)-based feature fusion module, the complementary interaction of low-level features and semantic fusion of deep features are realized. Liu Z et al. [18] proposed a cross-modal fusion model for GRB and thermal infrared salient target detection—SwinNet. Driven by the Swin Transformer, the method extracts hierarchical features. It bridges the gap between the two modalities driven by the attention mechanism to sharpen salient object contours guided by edge information. Xu F et al. [19, 20] considered that due to problems such as color cast and blur in underwater images, the features extracted directly from the backbone network often lack interesting and distinguishable features, which affects the performance of marine target detection. A novel exemplary ocean object detector based on an attention-based spatial pyramid pooling network and bidirectional feature fusion strategy is proposed to alleviate feature weakening and solve the ocean object detection problem. Then, a novel scale-aware feature pyramid structure SA-FPN is proposed to extract rich, robust features of underwater images and improve the performance of marine object detection. Wang H et al. [21, 22] aim at minimizing the reconstruction loss between input data and binary codes for autoencoder-based hashing algorithms while ignoring the potential consistency and complementarity of multi-source data, proposes an autoencoder-based multi-view binary clustering hashing algorithm that dynamically learns an associative graph with low-rank constraints, and employs collaborative learning between the autoencoder and the associative graph to learn a unified binary code. Then, considering that most existing methods have to introduce additional clustering steps to produce the final clusters, significantly reducing the unified relationship between graph learning and clustering, a multi-view clustering based on graph collaboration is proposed. Class Methods (MCGC). Xu F et al. [23] considered that the synthetic images are unrealistic enough, affecting the generalization to natural test images. They introduced segmentation masks to construct red, green, and blue mask pairs as input. They also designed an attention-guided style transfer network, learned style features from attention and background regions, and learned content features from entire and attention regions. The feature extraction process considers the target object's lower layers more. Semantics, but the interaction ability between high-level and low-level features is insufficient. At the same time, it

is challenging to balance high-level semantics and low-level features when establishing long-term dependencies. Therefore, we propose a local attention-guided Swin-transformer for thermal infrared sports object detection (LAGSwin) to address these limitations. In addition, our proposed thermal infrared moving target detection framework (LAGSwin) can be practically applied to thermal infrared imaging fault diagnosis [24] and other thermal infrared image target detection [25] tasks.

## Our proposed method

In this section, we first describe the overall architecture of the proposed LAGSwin detection framework; secondly, we elaborate on the feature extraction modules, namely the local attention-guided Swin-transformer component and the criss-cross PAFPN fusion component. Finally, the detection module with aligned convolutional filters is introduced, and the weighted loss function proposed in this paper is elaborated.

### Overview of the proposed LAGSwin

Contextual semantic information and spatial details are essential for target detection. Although the traditional global pooling operation can effectively aggregate this information, when the background semantics of the target is complex, or the target scale transformation is large, this operation may not be possible because the target information is unclear. Fully capture contextual, local, and global spatial details. Therefore, we propose the Local Attention Guided Swin-transformer [26–31] detection framework (LAGSwin) to address these limitations. The overall network architecture of LAGSwin is shown in Fig 1. The proposed LAGSwin detection framework is mainly composed of three essential modules: the local attention-guided Swin-transformer initial feature extraction module, the criss-cross fusion module (CCFM), and the ACDM. The initial feature extraction module comprises Swin-transformer and local attention guidance layer. Swin-transformer aims to explore the local features of sports targets and form feature maps of different scales at different stages. This multi-scale feature helps the network to cope with the target scale changes, and at the same time, the target can be described from different levels. The Local Attention Guidance Layer (LAG) makes up for the lack of low-level

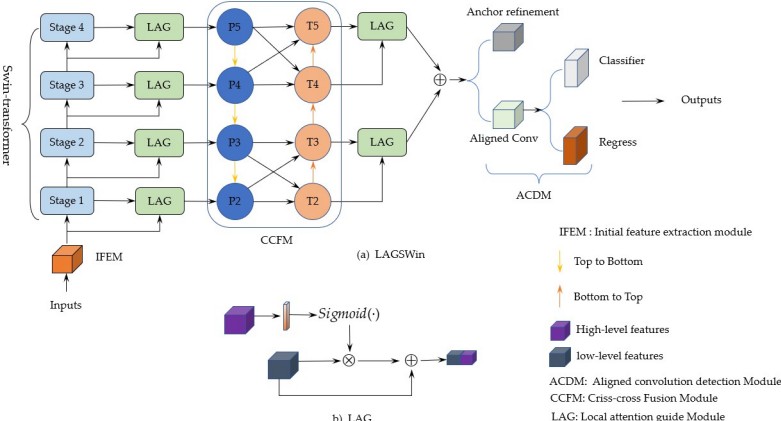

**Fig 1.** (a) The overall network structure of LAGSwin; (b) the network structure of the LAG module. Where ACDM represents the detection module of aligned convolution filters; CCFM represents the cross-fusion strategy; LAG represents the local attention guidance module; Stages 1 to 4 represent the four stages of Swin-transformer; P2 to P5 represent the top-down Pyramid structure features; T2 to T5 represent pyramid structure features from low to upper; ⊕ represent feature fusion; ⊗ represent matrix multiplication; *sigmoid*(·) represents activation function.

discriminative ability by embedding high-level features in low-level semantics and embedding some important low-level semantics in high-level features to make up for the high-level semantics in describing physics. The lack of basic attribute information, such as appearance, forms complementarity between high-level and low-level features, prompting the network to use the target's spatial details better. In contrast, CCFM uses top-down and bottom-up targets for modeling to fully capture the contextual semantics of the targets, which enables the proposed framework to establish more effective long-term dependencies on targets. It is worth noting that the criss-cross fusion strategy (CCFM) not only achieves further interaction between features at different levels but also further refines the spatial details and reduces the use of redundant information. In ACDM, an aligned convolution filter is introduced to ensure accurate encoding of orientation information and, simultaneously, to reduce the inconsistency between classification confidence and localization regression. In addition, we design a weighted loss function to act on the classification branch to make the network converge better and achieve more accurate classification.

## Local attention guided feature extraction module

This module is mainly composed of the Swin-transformer and Local Attention Guidance Layer (LAG), where the Swin-transformer is designed to explore the multi-scale local information of sports objects in thermal infrared images. At the same time, LAG is used for high-level and low-level features at different scales. Create interactions between them so there are dependencies between elements at different levels.

Assuming that the input thermal infrared image is $x \in R^{H \times W \times C}$, where, H, W, C represent the height, width and channel dimensions of the input feature, respectively, we use a set of dilated convolutions [32, 33] to perform initial feature extraction to obtain the initial feature f0; secondly, the feature maps generated by the four stages of Swin-transformer are $f_1, f_2, f_3$ and $f_4$ respectively. It is worth noting that in the initial feature extraction process, to maximize the preservation of global semantic information, we use a set of weighted pooling operations after each group of dilated convolutions, namely the mean of global average pooling and maximum pooling, preventing Overfitting while maximally preserving international semantic details. The calculation of the feature map $f_0$ is shown in Eq.

$$\begin{cases} z_{0,r} = DConv_r(x) \\ f_{0,r} = Mean(GAP(z_{0,r}), MaxPool(z_{0,r})) \\ f_0 = Conv_{1 \times 1}(Cat(f_{0,r})), r = 1, 2, 3 \end{cases} \tag{1}$$

Where, $r$ represents the expansion coefficient; $DConv(\cdot)$ represents the dilation convolution operation; $Conv_{1 \times 1}(\cdot)$ represents the convolution operation of $1 \times 1$; $Mean(\cdot)$ represents the matrix mean operation; $MaxPool(\cdot)$ represents the maximum pooling operation; $GAP(\cdot)$ represents the global average pooling operation; $Cat(\cdot)$ represents feature splicing;

To obtain better feature representation, at the same time, the complementarity between high-level features and low-level features is promoted. In addition, due to the differences in space and physical appearance of different targets, it isn't easy to form unified modeling. If the most common method is used to splice low-level features with high-level features, although it can bring a slight performance improvement, it will also cause redundant information. Therefore, to make full use of low-level features and, at the same time, to establish influential associations between them, we design a local attention guidance layer (LAG) under the condition of maximizing the preservation of spatial details and making up The gap between high-level and low-level features. The Local Attention Guidance Layer (LAG) operation on $f_0$ to $f_4$ is shown in

Eq.

$$\begin{cases} f_s' = LAG(\theta) \\ LAG(\theta) = f_{s-1} + f_s + \alpha_{s-1}f_{s-1} + \alpha_s f_s \\ s = 1, 2, 3, 4 \end{cases} \tag{2}$$

Where, $f_s'$ represents the local attention guidance features of different scales; $\alpha$ represents the local attention coefficient; $s$ represents the feature map scale; LAG represents the local attention guidance operation; $f_s$ represents high-level features; $f_{s-1}$ represents low-level features.

According to the equation, we can find that the operation of LAG is similar to residual connection in structure. When the expressive ability of low-level semantic information is better, it avoids excessive interference of high-level features, thus further emphasizing the importance of low-level features; that is to say, we designed this guiding method to help the representation of low-level semantic information, and at the same time, the expression of semantic information is enriched by the process of mutual embedding.

## Criss-cross fusion module (CCFM)

Feature Pyramid Network (FPN) [34–36] in the feature aggregation stage, pyramid feature maps of different scales are usually obtained along the bottom-up path. This way, when the shallow features are transferred to the top layer, they must go through multiple network layers, resulting in low-level features. Feature information is seriously lost. Therefore, to preserve the rich low-level semantic information to the greatest extent, we design a dual-path criss-cross fusion strategy, which encodes feature maps of different scales from bottom-up and top-down directions. The approach shortens the transmission path of low-level information flow between layers while ensuring the integrity and diversity of features. In addition, it further strengthens the interaction between high-level semantics and low-level semantic information. The fusion steps of CCFM are as follows.

First, the feature maps $f_s'$ obtained by the LAG module is output to the $1 \times 1$ convolutional layer for feature compression. The bottom-up path is used for the transfer to get pyramid feature maps $f_{P,s+1}$ of different scales, respectively (eg. $P2$ to $P5$ in CCFM in Fig 1). This features can be indicates as.

$$\begin{cases} f_{P,s+1} = Conv_{1\times1}(f_s') \\ s = 1, 2, 3, 4 \end{cases} \tag{3}$$

Where, $Conv_{1\times1}$ indicates the convolution operation of $1 \times 1$.

Secondly, the top-down path transfer is used to obtain pyramid features of different scales, and the criss-cross fusion strategy is used to get $f_{T,2}, f_{T,3}, f_{T,4}$ and $f_{T,5}$. The fusion process is shown in the equation.

$$\begin{cases} f_{T,2} = Conv_{1\times1}(Conv_{3\times3}(f_{P,2} \oplus f_{P,3})) \\ f_{T,3} = Conv_{1\times1}(Conv_{3\times3}(f_{P,2} \oplus f_{P,3} \oplus Conv_{1\times1}(f_{T,2}))) \\ f_{T,4} = Conv_{1\times1}(Conv_{3\times3}(Conv_{1\times1}(f_{T,3}) \oplus f_{P,4} \oplus f_{P,5})) \\ f_{T,5} = Conv_{1\times1}(Conv_{3\times3}(Conv_{1\times1}(f_{T,4}) \oplus f_{P,4} \oplus f_{P,5})) \end{cases} \tag{4}$$

Where, $Conv_{3\times3}(\cdot)$ represents the convolution operation of $3 \times 3$; $\oplus$ represents feature stitching; $Conv_{1\times1}$ represents the convolution operation of $1 \times 1$;

Although this dual-path criss-cross fusion strategy effectively captures multi-scale spatial details, it also guides some redundant information. Therefore, we again introduce the LAG

component to embed four different scales in each other to further improve the contextual semantics and global Representation performance of spatial details. The embedding process is shown in Eq.

$$\begin{cases} f'_{T,s+1} = f_{T,s-1} + f_{T,s} + \alpha_{T,s-1} f_{T,s-1} + \alpha_{T,s} f_{T,s} \\ s = 2, 4 \end{cases} \tag{5}$$

Where, $\alpha_{T,s-1}, \alpha_{T,s}$ represents the local attention map; it is worth noting that we fuse the features of these four different scales into two feature maps of high resolution and low resolution, namely high-level and low-level features information of $s = \{2, 4\}$.

In addition, we use a simple feature stitching method to stitch these two feature maps to obtain the feature map that is finally used to represent the target object. The fusion is shown in the equation.

$$f_o = UpSample(f_{T,5}) + f_{T,3} \tag{6}$$

Where, $UpSample(\cdot)$ indicates an upsampling operation.

## Detection module for aligned convolutional (ACDM)

This module includes aligned convolution filters, anchor refinement structure, and classification regression modules. Among them, the aligned convolution filter is aimed at the feature extraction module to obtain features for decoding. The anchor point refinement structure generates more accurate and higher quality anchor point boxes to improve classification accuracy. Simply put, these two components' primary purpose is to encode orientation information while reducing the inconsistency between classification and localization regression and enabling us to generate high-quality anchor and alignment features for accurately detecting sports objects in thermal infrared images.

To achieve the optimal performance of the proposed framework, we design a weighted loss function; the loss function is defined as the following equation.

$$\begin{cases} \zeta_{Total} = (\beta \zeta_{MCE}^h + \gamma \zeta_{MCE}^L) + \zeta_{FL} \\ \beta + \gamma = 1 \end{cases} \tag{7}$$

Where, $\zeta_{MCE}^h$ represents the loss function of high-level features; $\zeta_{MCE}^L$ represents the loss function of low-level features; $\zeta_{FL}$ represents the primary loss function of the proposed LSGSWin detection framework, which can effectively alleviate the problem of class imbalance; $\beta, \gamma$ represent weight factors;

**Algorithm 1**: The thermal infrared sports object detection process by our develop LAGSwin.

```
Input: Given an thermal infrared sports objects image x ∈ R^(H×W×C), the
backbone network of Swin-transformer generates feature maps f₁ to f₄
at different scales. Weighted overall loss function ζ_Total, the loss
function of high and low-level features is ζ^h_MCE and ζ^l_MCE, respectively.
The primary loss function ζ_FL.
for epochs = 0 to epochs = Max do
    f₀ ⟵^(IFEM) x ∈ R^(H×W×C);
    f'_s ⟵^(LAG) (f_s, f₀);
    f_(T,s+1) ⟵^(CCFM) f_(P,s+1) ⟵^(CCFM) f'_s;
    f'_(T,(3,5)) ⟵^(LAG) f_(T,s+1), s = {1, 2, 3, 4};
    Calculate finally features of f_o by Eq 6.;
```

```
   The output of LAGSwin is used for classification and
box regression.;
end
output:optimization the training during by AdamW and loss function of
ζTotal.
```

## Experimental results and analysis

To demonstrate the validity and reliability of the proposed LAGSwin detection framework, we use two open-source sports baseline data as experimental samples to evaluate the proposed method and the current, more advanced detection models. First, the data sources are introduced in detail, and the evaluation metrics and parameters are given; second, the ablation research and analysis discussions are presented.

### Data preparation

**TTsports** [16]. This data set is captured by a Q1922-type thermal camera. There are a total of 4 30-second indoor football sequences. For the consistency of the experiment, we rearranged the data set and generated a total of 1500 images after processing. 1920 × 480 size image, each thermal infrared image contains eight different sports players.

**FLIRs** [37]. This dataset was released in July 2018, with a total of 14, 000 images, 10, 000 of which are from short video clips, and another 4, 000 BONUS images are from a 140-second video. These thermal infrared images mainly include four categories of people, cars, dogs, and other vehicles.

**RGBsports** [16]. This dataset contains 3000 RGB images, including 1874 footballs and 1126 crickets. For the fairness of the experiment, we randomly selected 40% of them as training samples, 10% as verification samples, and the remaining 50% were used to evaluate all detection models. It is worth noting that in data set processing, we use the overlap ratio method for cropping for data set expansion. At the same time, we use different ratios to adjust the scaling ratio. The scaling ratio parameters are set to 0.5, 1.0, and 1.5.

### Parameter settings and evaluation Metrics

We adopt AdamW as the optimizer to tune and optimize the whole detection framework, where the learning rate is set to 0.0025, the number of iterations is set to 36, and the batch size is set to 16. Meanwhile, the pre-training of ImgaeNet22k is used to obtain better feature representation. The weights are parametrically tuned to the Swin-transformer module. In the training process, we adopted two training strategies, single-scale and multi-scale, in which the multi-scale size was set to 600 × 480 and 600 × 800, and the scale of the test phase was 600 × 600.

To ensure the smooth progress of the experiments, all experiments were completed on 4 RTX3090 of python3.7.6 and torch1.7.0+cu110, and the recall rate and mean average precision rate (mAP) were used as evaluation indicators. The calculation process is shown in the equation.

$$\begin{cases} Recall = \dfrac{TP}{TP + FN} \\[2mm] Precision = \dfrac{TP}{TP + FP} \\[2mm] mAP = \dfrac{1}{N} \displaystyle\int_0^1 Precision(R) \end{cases} \tag{8}$$

Where TP (True Positive) indicates that a detection frame with an intersection ratio ($IoU$) > 0.5 with the Ground Truth target frame is detected, it is worth noting that the same Ground Truth is only calculated once. FP (False Positive) indicates the number of detection boxes with the target box $IoU$ < = 0.5 or the number of redundant detection boxes with the same Ground Truth detected. FN (False Negative) represents the number of target boxes that are not seen. mAP (mean Average Precision) represents the average value of each category of AP, and AP is to calculate the area under the P-R curve of a specific type. The larger the mapped value, the better the detection effect of the method.

## Comparison with advanced methods

To demonstrate the effectiveness of the proposed LAGSwin detection framework, we conduct experimental comparisons on multiple detection models; Table 1 presents the experimental results of different methods. From Table 1, we can draw the following conclusions:

(1) The thermal infrared sports target detection framework of LAGSwin we proposed has achieved the best detection performance on three open-source datasets, including TTsports, RGBsports, and FLIRs. For example, the mAP on the TTsports, RGBsports, and FLIRs data sets is 0.057, 0.017, and 0.025 higher than the ReDet method. The possible reason is that, on the one hand, we describe the targets in thermal infrared images in detail from three different scales and levels of low, medium, and high, and establish functional spatial and long-term dependencies between these features and use The local attention guidance layer highlights the details, allowing the network to better focus on the subtle changes between different types of objects, as well as the differences between objects and backgrounds. On the other hand, introducing convolutional filters in the detector, encoding orientation information while reducing the inconsistency between classification and localization regression, enables us to generate high-quality anchor and alignment features. In addition, each component assists the network in obtaining the optimal feature representation, which ultimately leads to the optimal performance of the proposed model on the three datasets.

(2) Compared with single-stage detection methods such as SASM, Oriented RepPoints, and KLD, two-stage detection methods such as ReDet, Roi-transformer, and Oriented R-CNN have achieved strong competitiveness on these open source datasets. For example, the mAP of the Roi-transformer is 0.079, 0.039, and 0.039 higher than the Oriented RepPoints detection method, respectively. The mAP of Oriented R-CNN is 0.06, 0.068, and 0.055, higher

**Table 1. Experimental results of different detection methods.** "FLOPs" indicates the number of floating-point operations per second, and the unit is *GM*; "Parameter" indicates the number of parameters, and the unit is ×10*M*.

| Models | TTsports | | RGBsports | | FLIRs | | FLOPs | Parameters |
|---|---|---|---|---|---|---|---|---|
| – | R | mAP | R | mAP | R | mAP | – | – |
| ReDet | 0.942 | 0.797 | 0.909 | 0.818 | 0.891 | 0.802 | 11.4 | 9.5 |
| Roi-transformer | 0.935 | 0.788 | 0.879 | 0.760 | 0.861 | 0.744 | 14.6 | 8.6 |
| Oriented R-CNN | 0.914 | 0.738 | 0.798 | 0.695 | 0.774 | 0.669 | 15.8 | 8.2 |
| DiffusionDet(2023) | 0.933 | 0.791 | 0.902 | 0.809 | 0.885 | 0.799 | 12.5 | 9.1 |
| Oriented RepPoints | 0.904 | 0.709 | 0.870 | 0.721 | 0.854 | 0.705 | 21.2 | 5.7 |
| SASM | 0.839 | 0.621 | 0.727 | 0.545 | 0.704 | 0.521 | 21.8 | 3.3 |
| KLD | 0.873 | 0.678 | 0.855 | 0.627 | 0.817 | 0.614 | 20.7 | 4.9 |
| Rtmdet(2022) | 0.947 | 0.821 | 0.911 | 0.794 | 0.896 | 0.815 | 22.3 | 5.2 |
| **LAGSwin** | **0.955** | **0.854** | **0.929** | **0.835** | **0.909** | **0.827** | **19.4** | **6.6** |

than that of the KLD detection method. It may be that the two-stage detection method generates high-quality regions of interest in the first stage, which prompts the network to learn a better feature representation. In addition, the SASM detection method performed the worst on the three open-source datasets. The possible reason is that the SASM method focuses on the representation of object shape information while ignoring the extraction of high-level discriminative semantics. It is worth noting that the single-stage detection method Oriented RepPoints has achieved better competitiveness on the RGBsports and FLIRs datasets. For example, the mAP of Oriented RepPoints is 0.026 and 0.036 higher than Oriented R-CNN, respectively. It is possible that the Oriented RepPoints detection method uses adaptive point representation and dynamic evaluation and allocation strategies, which promotes the network to capture any instance-oriented geometric information effectively, and uses three orientation conversion functions to achieve accurate positioning of the target, and at the same time filters the feature point set Highlighting the representation improves the classification accuracy, which finally leads to the Oriented RepPoints detection method outperforming the Oriented R-CNN method. In addition, the current advanced single-stage detection algorithm Rtmdet and two-stage detection algorithm DiffusionDet have achieved good competitive advantages in the three datasets, such as mAP on the FLIRs data set are 0.815 and 0.799, respectively. The detection efficiency of Rtmdet is better than DiffusionDet.

(3) The proposed LAGSwin detection framework is still highly competitive in reasoning efficiency while ensuring optimal detection accuracy. For example, the FLOPs of LAGSwin are 1.8, 2.4, and 1.3 lower than single-stage detection methods such as Oriented RepPoints, SASM, and KLD, respectively. Still, our detection accuracy is far superior to these methods. Compared with the two-stage detection method, our proposed LAGSwin detection framework achieves the best detection accuracy and inference efficiency (FLOPs).

## Ablation studies

**The component of LAGSwin framework**. We use quantitative and qualitative methods to verify each part to prove whether each component in the proposed LAGSwin detection framework plays a positive role in the model. Table 2 presents the experimental results in different detail. From Table 2, we draw the following conclusions:

(1) In our proposed LAGSwin thermal infrared sports object detection framework, each component plays a crucial positive role in the overall performance of the framework. On the three open source data sets, the backbone network using PVTV2 as the model has achieved a tremendous competitive advantage, such as the mAP of Resnet101+DCN increased by 0.01, 0.01, and 0.001, respectively. On the TTsports and FLIRs data sets, compared with Res2net101 mAP increased by 0.002 and 0.004, respectively, but decreased by 0.006 on the RGBsports dataset. The main reason is that PVTV2 benefits from the self-attention mechanism in Transformer and always maintains the global receptive field, ensuring the local semantics of the target and better acquisition of the target. The global details of the RGBsports data set may be reduced because the target scale changes significantly or the target background is more complex, which reduces the detection performance of PVTV2. In addition, HRNet obtained the worst detection performance, which was 0.007, 0.009, and 0.002 lower than MobileNetV2 on the three open-source datasets. The possible reason is that HRNet focuses on obtaining high-resolution feature information of the target, ignoring the rich low-resolution feature information. Hierarchical semantic information also shows that low-level semantic information helps detect thermal infrared sports targets.

**Table 2. Experimental results of different component.** IFEM+Backbone+CCFM+ACDM demonstrates that the model does not use any local attention guidance layer; IFEM+Backbone+LAG+FPN+ACDM indicates that the FPN is used to replace the CCFM component; IFEM+Backbone+LAG+FPN+ACDM indicates that the PAFPN is used to replace the CCFM component. 'Backbone' indicates the feature extractor of SWin-transformer.

| Components | TTsports | | RGBsports | | FLIRs | |
|---|---|---|---|---|---|---|
| – | R | mAP | R | mAP | R | mAP |
| Resnet50 | 0.947 | 0.843 | 0.924 | 0.817 | 0.902 | 0.808 |
| Resnet101 | 0.937 | 0.841 | 0.906 | 0.828 | 0.897 | 0.811 |
| Resnet50+DCN | 0.948 | 0.845 | 0.926 | 0.819 | 0.905 | 0.812 |
| Resnet101+DCN | 0.950 | 0.842 | 0.919 | 0.829 | 0.904 | 0.820 |
| HRNet | 0.908 | 0.829 | 0.904 | 0.806 | 0.889 | 0.799 |
| PVTV2 | 0.952 | 0.852 | 0.926 | 0.833 | 0.907 | 0.821 |
| Resnest50 | 0.947 | 0.840 | 0.919 | 0.814 | 0.890 | 0.802 |
| Resnest101 | 0.945 | 0.847 | 0.913 | 0.828 | 0.901 | 0.809 |
| MobileNetV2 | 0.914 | 0.836 | 0.896 | 0.815 | 0.900 | 0.801 |
| Res2net50 | 0.943 | 0.844 | 0.921 | 0.819 | 0.891 | 0.805 |
| Res2net101 | 0.951 | 0.850 | 0.920 | 0.839 | 0.902 | 0.817 |
| IFEM+Backbone+LAG+ACDM | 0.901 | 0.808 | 0.900 | 0.802 | 0.891 | 0.805 |
| IFEM+Backbone+CCFM+ACDM | 0.908 | 0.821 | 0.898 | 0.801 | 0.897 | 0.813 |
| IFEM+Backbone+LAG+FPN+ACDM | 0.955 | 0.851 | 0.919 | 0.830 | 0.904 | 0.822 |
| IFEM+Backbone+LAG+PAFPN+ACDM | 0.952 | 0.848 | 0.917 | 0.832 | 0.912 | 0.825 |
| IFEM+Backbone+LAG+HRFPN+ACDM | 0.949 | 0.843 | 0.911 | 0.829 | 0.900 | 0.821 |
| Backbone+LAG+CCFM+ACDM | 0.950 | 0.846 | 0.914 | 0.831 | 0.908 | 0.822 |

(2) Compared with the external feature extraction network, the deep backbone network can better capture the features of thermal infrared sports targets. For example, on the FLIRs dataset, the mAP of Res2net101, Resnet101, and Resnest101 are 0.012, 0.007, and 0.003 higher than Res2net50, Resnet50, and Resnest50, respectively. Similarly, on the RGBsports and FLIRs datasets, the mAP of Resnet101+DCN is higher than Resnet50+DCN. 0.01 and 0.008. The possible reason is that as the number of network layers deepens, the deep backbone network acquires more distinguishable high-level features, highlighting the differences between different types of targets. Compared with Resnet and Resnest backbone networks, the overall performance of Res2net has strong competitiveness. For example, on the TTsports dataset, the mAP of Res2net50 is 0.001 and 0.004 higher than Resnet50 and Resnest50, respectively. The same R-value performs poorly. In addition, as the number of network layers deepens, the detection accuracy of all data has improved. But on the RGBsports data, the mAP of Resnet101 and Resnest101 are equal.

(3) Using Res2net101 as the backbone network to replace the Swin-transformer in our model has achieved the best competitiveness. At the same time, on the RGBsports data, the mAP is 0.004 higher than that of the Swin-transformer. This may be because, in the local feature capture stage, res2net101 first divides the target features in the RGB image into multiple subspaces so that the network can obtain more detailed local semantics. Still, the R value is low, which may be the model with the deepening of the number of network layers leads to the utilization of a large amount of redundant information, which weakens the representation of the global semantics of the target. The model may also have fallen into a local optimum, and overfitting has occurred.

(4) IFEM+Backbone+CCFM+ACDM showed the worst performance on TTsports, RGBsports, and FLIRs data sets. The possible reason is significant spatial distribution and

**Table 3. Experimental results of different loss function.** $\zeta_{MCE}$ indicates that only cross-entropy loss is used; $\zeta_{FL}$ indicates that only Focal loss is used to adjust and optimize the entire network; $\zeta_{MCE} + \zeta_{FL}$ indicates that simple weighted loss is used, that is, cross-entropy and Focal loss are used together to act on the network.

| Components | TTsports | | RGBsports | | FLIRs | |
|---|---|---|---|---|---|---|
| – | R | mAP | R | mAP | R | mAP |
| $\zeta_{MCE}$ | 0.952 | 0.851 | 0.908 | 0.820 | 0.910 | 0.815 |
| $\zeta_{FL}$ | 0.954 | 0.855 | 0.927 | 0.833 | 0.918 | 0.825 |
| $\zeta_{MCE} + \zeta_{FL}$ | 0.954 | 0.853 | 0.919 | 0.831 | 0.914 | 0.822 |

physical meaning differences between different scales. If you directly use the simple fusion method, It is challenging to balance the differences between them. Still, the LAG layer we designed plays an essential role using the weight distribution and residual strategy. IFEM +Backbone+LAG+PAFPN+ACDM has gained a better competitive advantage than IFEM +Backbone+LAG+FPN+ACDM. This may be because PAFPN also uses its own. The top-down and bottom-up two-way information transmission strategy reduces the loss of details caused by information transmission while preserving the global semantics to the greatest extent. In addition, the IFEM+Backbone+LAG+ACDM method achieved the worst performance on the three sets of data sets, with mAP of 0.808, 0.802, and 0.805, respectively, which shows that CCFM plays a positive role in the overall framework and is beneficial to the contextual detail features. The Backbone+LAG+CCFM+ACDM method has achieved better competitiveness. At the same time, it also shows that IFEM benefits the representation of prior knowledge.

**Loss function**. To demonstrate that our proposed weighted loss function has a positive effect on the overall performance of the model, different loss functions are used to test on TTsports, RGBsports and FLIRs datasets. The experimental results are shown in Table 3.

From Table 3, we can find that when $\zeta_{FL}$ is used to optimize and adjust the proposed detection framework, the detection performance has strong competitiveness in three groups of open-source datasets, such as TTsports, RGBsports, and FLIRs. For example, the mAP of $\zeta_{FL}$ is 0.002, 0.002, and 0.003 higher than that of $\zeta_{MCE} + \zeta_{FL}$. The possible reason is an imbalance in the target categories in these data. The $\zeta_{FL}$ loss function can effectively deal with the imbalance of categories, thus obtaining the best detection performance. Compared with the other two open-source datasets, the mce loss function performs the worst in RGBsports. It may be that the size of the target category in this data is small, and the scale change between different categories is small, as well as the existence of a category imbalance problem degrades the final detection performance.

## Discussion

To visually demonstrate the effectiveness of the proposed LAGSwin detection framework, Fig 2 and Table 4 show the detection performance of the model for each category on different datasets of TTsports, RGBsports and FLIRs.

According to Table 4 we can find:

(1) In the RGBsports data set, compared to the baseball class whose ID number is 0, the detection effect of football is significantly better than that of baseball, namely, AP and R have increased by 0.184 and 0.116, respectively. The possible reason is that the shape and size of the football are easy. Distinguishing from the image background enables the network to learn a more practical difference between the target and the background, thereby improving

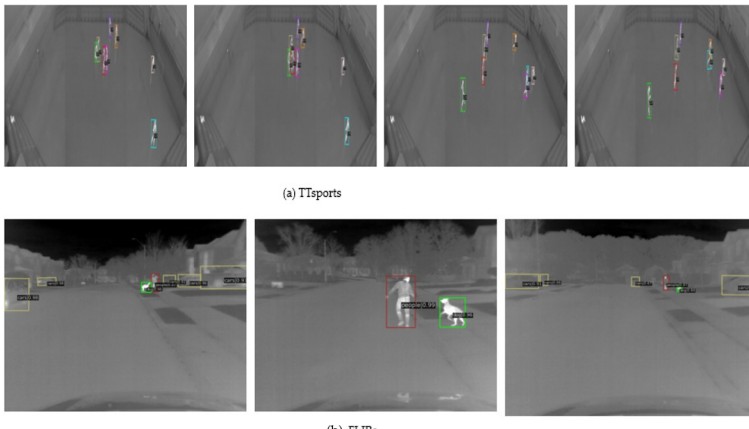

**Fig 2. The detection effect of ours model on different dataset.** (a) indicates the datasets of TTsports. (b) indicates the datasets of FLIRs. Different color detection boxes in the image represent different object classes.

the detection performance of football. The smaller size of the baseball is easy to confuse with the target background and causes misclassification. In the FLIRs data, the *otherscars* class with ID number 3 achieved the worst performance, probably because this class has a small number of samples, and the slight difference between its *cars* caused misclassification. This reduces the detection accuracy and the performance of the detection framework for the *cars* class. For example, the APs of *dog* and *people* are 0.053 and 0.069 higher than *cars*, respectively.

(2) In the TTsports dataset, the detection accuracy of each class is very competitive. It is worth noting that although *sportsman*6 and *sportsman*3 have the same detection performance, AP is 0.838, but R is 0.919 and 0.925, respectively. The possible reason is that *sportsman*6 has occlusion during the movement, which reduces the R-value. A visual presentation of the different data is shown in Fig 2.

## Conclusions and next research

In this paper, we propose a local attention-guided swain-transformer detection framework (LAGSwin) for detecting spatial details of moving objects in thermal infrared images. The

**Table 4. Experimental results for each category on different datasets of TTsports, RGBsports and FLIRs.** "Classes" indicates the object class in this datasets. "AP" represents the average detection accuracy of the objects. "ClassID" indicates the ID number of the category in the dataset.

| ClassID | TTsports | | | FLIRs | | | RGBsports | | |
|---|---|---|---|---|---|---|---|---|---|
| – | Classes | R | AP | Classes | R | AP | Classes | R | AP |
| 0 | sportsman1 | 0.976 | 0.895 | people | 0.976 | 0.884 | baseball | 0.871 | 0.761 |
| 1 | sportsman2 | 0.913 | 0.835 | cars | 0.912 | 0.815 | football | 0.987 | 0.909 |
| 2 | sportsman3 | 0.925 | 0.838 | dog | 0.934 | 0.868 | – | – | – |
| 3 | sportsman4 | 0.974 | 0.868 | otherscars | 0.814 | 0.741 | – | – | – |
| 4 | sportsman5 | 0.969 | 0.865 | – | – | – | – | – | – |
| 5 | sportsman6 | 0.919 | 0.838 | – | – | – | – | – | – |
| 6 | sportsman7 | 0.832 | 0.844 | – | – | – | – | – | – |
| 7 | sportsman8 | 0.943 | 0.849 | – | – | – | – | – | – |

method first uses a feature extraction module guided by local attention to strengthen the interaction between low-level and high-level features. It embeds high-level features into low-level features so that high-level features contain rich low-level semantics. Embedding high-level features into low-level features makes low-level features more discriminative in high-level semantics. Secondly, design a cross-fusion strategy to aggregate these feature information from different directions, reduce redundant information while retaining spatial details to the greatest extent, and ensure the integrity and diversity of attribute information such as the physical appearance of the target; in the detection module, complete The feature alignment of the algorithm alleviates the inconsistency between regression and classification. Finally, evaluation tests were performed on three sets of open-source baseline data, including TTsports, FLIRs, and RGBsports, and optimal detection performance and good robustness were achieved.

During the experiment, we found that the design of the feature extraction module of the detection framework is relatively complicated, which increases the redundancy of the model. At the same time, the detection efficiency also has a lot of room for improvement. Therefore, in the following research, we will start from the above two aspects to design a simple and efficient semantic guidance network, that is, to lighten the feature extractor and design a more effective semantic fusion module., to preserve the spatial details of the target in the thermal infrared image to the greatest extent, and at the same time, use the new semantic fusion module to efficiently gather different levels of semantics to improve feature representation performance.

## Author Contributions

**Formal analysis:** Hengran Meng.

**Investigation:** Hengran Meng, Shuqi Si.

**Methodology:** Shuqi Si.

**Resources:** Liping Wu.

**Software:** Shuqi Si, Liping Wu.

**Writing – original draft:** Jia Zhao.

**Writing – review & editing:** Bingfei Mao.

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
