## [Decision Letter · Decision Letter 0]

3 Jul 2023

PONE-D-23-13636LAGSwin: Local attention guided Swin-transformer for thermal infrared sports object detectionPLOS ONE

  Dear Dr. Mao,

Thank you for submitting your manuscript to PLOS ONE. After careful consideration, we feel that it has merit but does not fully meet PLOS ONE’s publication criteria as it currently stands. Therefore, we invite you to submit a revised version of the manuscript that addresses the points raised during the review process.

ACADEMIC EDITOR:Please consider and respond all comments from the reviewers very carefully before submitting your revised version.

We look forward to receiving your revised manuscript.

Kind regards,

Nattapol Aunsri, Ph.D.

Academic Editor

PLOS ONE

3. We note that Figure 2 in your submission contain copyrighted images. All PLOS content is published under the Creative Commons Attribution License (CC BY 4.0), which means that the manuscript, images, and Supporting Information files will be freely available online, and any third party is permitted to access, download, copy, distribute, and use these materials in any way, even commercially, with proper attribution. For more information, see our copyright guidelines: http://journals.plos.org/plosone/s/licenses-and-copyright.

b.If you are unable to obtain permission from the original copyright holder to publish these figures under the CC BY 4.0 license or if the copyright holder’s requirements are incompatible with the CC BY 4.0 license, please either i) remove the figure or ii) supply a replacement figure that complies with the CC BY 4.0 license. Please check copyright information on all replacement figures and update the figure caption with source information. If applicable, please specify in the figure caption text when a figure is similar but not identical to the original image and is therefore for illustrative purposes only.

Reviewers' comments:

Reviewer's Responses to Questions

**Comments to the Author**

1. Is the manuscript technically sound, and do the data support the conclusions?

Reviewer #1: Yes

Reviewer #2: No

2. Has the statistical analysis been performed appropriately and rigorously? 

Reviewer #1: Yes

Reviewer #2: Yes

3. Have the authors made all data underlying the findings in their manuscript fully available?

Reviewer #1: Yes

Reviewer #2: Yes

4. Is the manuscript presented in an intelligible fashion and written in standard English?

Reviewer #1: Yes

Reviewer #2: Yes

5. Review Comments to the Author

Reviewer #1: -The paper should be interesting ;;;

-it is a good idea to add more photos of measurements, sensors + arrows/labels what is what (if any);;;

-it is a good idea to add a block diagram of the proposed research (step by step);;;

-too many formulas;;;

-What is the result of the analysis?;;

-figures should have high quality. ;;;;;

-text should be formatted;;;;

-please add photos of the application of the proposed research, 2-3 photos ;;;

-what will society have from the paper?;;

-Is there a possibility to use the proposed research for other topics, neural networks, image processing etc.;;;

"Thermographic fault diagnosis of electrical faults of commutator and induction motors";;;

"Thermographic Fault Diagnosis of Shaft of BLDC Motor";;;

-references should be from the web of science 2020-2023 (50% of all references, 30 references at least);;;

-please compare advantages/disadvantages of other approaches;;;

-Conclusion: point out what have you done;;;;

-please add some sentences about future work;;;

Reviewer #2: This manuscript proposed a local attention-guided Swin-transformer thermal infrared sports object detection method (LAGSwin) to encode sports objects’ spatial transformation and orientation information. The evaluation and verification of multiple open-source sports target datasets prove that the proposed LAGSwin detection framework has good robustness and generalization ability. However, there are some concerns about this manuscript:

1: In related work, the author introduced the one-stage detector and the two-stage detector. It is recommended that the transformer detector be further supplemented.

2: In ablation experiment, the ablation results of local attention guided feature extraction module, criss-cross fusion module and detection module for aligned convolutional should be provided to justify the detailed design.

3: Subjective experiment content is insufficient, which needs to be further supplemented to improve the credibility of the work.

4: In comparison experiments, it is recommended to provide fair comparison results with the state of the art schemes.

5: For the task of object detection and attention, some recent works should be discussed, including "Refined marine object detector with attention-based spatial pyramid pooling networks and bidirectional feature fusion strategy", "Scale-aware feature pyramid architecture for marine object detection", "Graph-Collaborated Auto-Encoder Hashing for Multi-view Binary Clustering" and "Towards Adaptive Consensus Graph: Multi-view Clustering via Graph Collaboration", "Purifying real images with an attention-guided style transfer network for gaze estimation".

6: We would like to know the comparison results (accuracy, parameters, computations) of the proposed algorithm with the current popular one-stage, two-stage and transformer detectors.

6. PLOS authors have the option to publish the peer review history of their article (what does this mean?). If published, this will include your full peer review and any attached files.

Reviewer #1: No

Reviewer #2: No

---

## [Author Response · Author response to Decision Letter 0]

25 Sep 2023

Reviewer#1, Concern # 1: it is a good idea to add more photos of measurements, sensors + arrows/labels what is what (if any).

Author response: Thank you very much for the expert’s opinion. I have modified this paper in detail in response to this opinion.

Author action: We updated the manuscript by. 

According to Table 4 we can find:

(1) In the RGBsports data set, compared to the baseball class whose ID number is 0, the detection effect of football is significantly better than that of baseball; that is, AP and R have increased by 0.184 and 0.116, respectively. The possible reason is that the shape and size of the football are easy. Distinguishing from the image background enables the network to learn a more practical difference between the target and the background, thereby improving the detection performance of football. The smaller size of the baseball is easy to confuse with the target background and causes misclassification. In the FLIRs data, the "otherscars" class with ID number 3 achieved the worst performance, probably because this class has a small number of samples, and the slight difference between its "cars" caused misclassification. This reduces the detection accuracy and the performance of the detection framework for the "cars" class. For example, the APs of "dog" and "people" are 0.053 and 0.069 higher than "cars," respectively.

(2) In the TTsports dataset, the detection accuracy of each class is very competitive. It is worth noting that although "sportsman6" and "sportsman3" have the same detection performance, AP is ${0.838}$, but R is 0.919 and 0.925, respectively. The possible reason is that "sportsman6" has occlusion during the movement, which reduces the R-value. A visual presentation of the different data is shown in Figure 2. 

Reviewer#1, Concern # 2: it is a good idea to add a block diagram of the proposed research (step by step)

Author response: Thank you very much for the expert’s opinion. I have modified this paper in detail in response to this opinion.

Author action: We updated the manuscript by 

Contextual semantic information and spatial details are essential for target detection. Although the traditional global pooling operation can effectively aggregate this information, when the background semantics of the target is complex, or the target scale transformation is large, this operation may not be possible because the target information is unclear. Fully capture contextual, local, and global spatial details. Therefore, we propose the Local Attention Guided Swin-transformer detection framework (LAGSwin) to address these limitations. The overall network architecture of LAGSwin is shown in Figure 1. The proposed LAGSwin detection framework is mainly composed of three essential modules: the local attention-guided Swin-transformer initial feature extraction module, the criss-cross fusion module (CCFM), and the ACDM. The initial feature extraction module comprises Swin-transformer and local attention guidance layer. Swin-transformer aims to explore the local features of sports targets and form feature maps of different scales at different stages. This multi-scale feature helps the network to cope with the target scale changes, and at the same time, the target can be described from different levels. The Local Attention Guidance Layer (LAG) makes up for the lack of low-level discriminative ability by embedding high-level features in low-level semantics and embedding some important low-level semantics in high-level features to make up for the high-level semantics in describing physics. The lack of basic attribute information, such as appearance, forms complementarity between high-level and low-level features, prompting the network to use the target's spatial details better. In contrast, CCFM uses top-down and bottom-up targets for modeling to fully capture the contextual semantics of the targets, which enables the proposed framework to establish more effective long-term dependencies on targets. It is worth noting that the criss-cross fusion strategy (CCFM) not only achieves further interaction between features at different levels but also further refines the spatial details and reduces the use of redundant information. In ACDM, an aligned convolution filter is introduced to ensure accurate encoding of orientation information and, simultaneously, to reduce the inconsistency between classification confidence and localization regression. In addition, we design a weighted loss function to act on the classification branch to make the network converge better and achieve more accurate classification.

Reviewer#1, Concern # 3: too many formulas; What is the result of the analysis?

Author response: Thank you very much for the expert’s opinion. I have modified this paper in detail in response to this opinion.

Author action: We updated the manuscript by. 

From Table 1, we can draw the following conclusions:

(1) The thermal infrared sports target detection framework of LAGSwin we proposed has achieved the best detection performance on three open-source datasets, including TTsports, RGBsports, and FLIRs. For example, the mAP on the TTsports, RGBsports, and FLIRs data sets is 0.057, 0.017, and 0.025 higher than the ReDet method. The possible reason is that, on the one hand, we describe the targets in thermal infrared images in detail from three different scales and levels of low, medium, and high, and establish functional spatial and long-term dependencies between these features and use The local attention guidance layer highlights the details, allowing the network to better focus on the subtle changes between different types of objects, as well as the differences between objects and backgrounds. On the other hand, introducing convolutional filters in the detector, encoding orientation information while reducing the inconsistency between classification and localization regression, enables us to generate high-quality anchor and alignment features. In addition, each component assists the network in obtaining the optimal feature representation, which ultimately leads to the optimal performance of the proposed model on the three datasets.

(2) Compared with single-stage detection methods such as SASM, Oriented RepPoints, and KLD, two-stage detection methods such as ReDet, Roi-transformer, and Oriented R-CNN have achieved strong competitiveness on these open-source datasets. For example, the mAP of the Roi-transformer is 0.079, 0.039, and 0.039 higher than the Oriented RepPoints detection method, respectively. The mAP of Oriented R-CNN is 0.06, 0.068, and 0.055, higher than that of the KLD detection method. It may be that the two-stage detection method generates high-quality regions of interest in the first stage, which prompts the network to learn a better feature representation. In addition, the SASM detection method performed the worst on the three open-source datasets. The possible reason is that the SASM method focuses on the representation of object shape information while ignoring the extraction of high-level discriminative semantics. It is worth noting that the single-stage detection method Oriented RepPoints has achieved better competitiveness on the RGBsports and FLIRs datasets. For example, the mAP of Oriented RepPoints is 0.026 and 0.036 higher than Oriented R-CNN, respectively. It is possible that the Oriented RepPoints detection method uses adaptive point representation and dynamic evaluation and allocation strategies, which promotes the network to capture any instance-oriented geometric information effectively, and uses three orientation conversion functions to achieve accurate positioning of the target, and at the same time filters the feature point set Highlighting the representation improves the classification accuracy, which finally leads to the Oriented RepPoints detection method outperforming the Oriented R-CNN method. In addition, the current advanced single-stage detection algorithm Rtmdet and two-stage detection algorithm DiffusionDet have achieved good competitive advantages in the three datasets, such as mAP on the FLIRs data set are 0.815 and 0.799, respectively. The detection efficiency of Rtmdet is better than DiffusionDet.

(3) The proposed LAGSwin detection framework is still highly competitive in reasoning efficiency while ensuring optimal detection accuracy. For example, the FLOPs of LAGSwin are 1.8, 2.4, and 1.3 lower than single-stage detection methods such as Oriented RepPoints, SASM, and KLD, respectively. Still, our detection accuracy is far superior to these methods. Compared with the two-stage detection method, our proposed LAGSwin detection framework achieves the best detection accuracy and inference efficiency (FLOPs).

From Table 2, we draw the following conclusions:

(1) In our proposed LAGSwin thermal infrared sports object detection framework, each component plays a crucial positive role in the overall performance of the framework. On the three open-source data sets, the backbone network using PVTV2 as the model has achieved a tremendous competitive advantage, such as the mAP of Resnet101+DCN increased by 0.01, 0.01, and 0.001, respectively. On the TTsports and FLIRs data sets, compared with Res2net101 mAP increased by 0.002 and 0.004, respectively, but decreased by 0.006 on the RGBsports dataset. The main reason is that PVTV2 benefits from the self-attention mechanism in Transformer and always maintains the global receptive field, ensuring the local semantics of the target and better acquisition of the target. The global details of the RGBsports data set may be reduced because the target scale changes significantly or the target background is more complex, which reduces the detection performance of PVTV2. In addition, HRNet obtained the worst detection performance, which was 0.007, 0.009, and 0.002 lower than MobileNetV2 on the three open-source datasets. The possible reason is that HRNet focuses on obtaining high-resolution feature information of the target, ignoring the rich low-resolution feature information. Hierarchical semantic information also shows that low-level semantic information helps detect thermal infrared sports targets.

(2) Compared with the external feature extraction network, the deep backbone network can better capture the features of thermal infrared sports targets. For example, on the FLIRs dataset, the mAP of Res2net101, Resnet101, and Resnest101 are 0.012, 0.007, and 0.003 higher than Res2net50, Resnet50, and Resnest50, respectively. Similarly, on the RGBsports and FLIRs datasets, the mAP of Resnet101+DCN is higher than Resnet50+DCN. 0.01 and 0.008. The possible reason is that as the number of network layers deepens, the deep backbone network acquires more distinguishable high-level features, highlighting the differences between different types of targets. Compared with Resnet and Resnest backbone networks, the overall performance of Res2net has strong competitiveness. For example, on the TTsports dataset, the mAP of Res2net50 is 0.001 and 0.004 higher than Resnet50 and Resnest50, respectively. The same R-value performs poorly. In addition, as the number of network layers deepens, the detection accuracy of all data has improved. But on the RGBsports data, the mAP of Resnet101 and Resnest101 are equal.

(3) Using Res2net101 as the backbone network to replace the Swin-transformer in our model has achieved the best competitiveness. At the same time, on the RGBsports data, the mAP is 0.004 higher than that of the Swin-transformer. This may be because, in the local feature capture stage, res2net101 first divides the target features in the RGB image into multiple subspaces so that the network can obtain more detailed local semantics. Still, the R value is low, which may be the model with the deepening of the number of network layers leads to the utilization of a large amount of redundant information, which weakens the representation of the global semantics of the target. The model may also have fallen into a local optimum, and overfitting has occurred. 

(4) IFEM+Backbone+CCFM+ACDM showed the worst performance on TTsports, RGBsports, and FLIRs data sets. The possible reason is significant spatial distribution and physical meaning differences between different scales. If you directly use the simple fusion method, It is challenging to balance the differences between them. Still, the LAG layer we designed plays an essential role using the weight distribution and residual strategy. IFEM+Backbone+LAG+PAFPN+ACDM has gained a better competitive advantage than IFEM+Backbone+LAG+FPN+ACDM. This may be because PAFPN also uses its own. The top-down and bottom-up two-way information transmission strategy reduces the loss of details caused by information transmission while preserving the global semantics to the greatest extent. In addition, the IFEM+Backbone+LAG+ACDM method achieved the worst performance on the three sets of data sets, with mAP of 0.808, 0.802, and 0.805, respectively, which shows that CCFM plays a positive role in the overall framework and is beneficial to the contextual detail features. The Backbone+LAG+CCFM+ACDM method has achieved better competitiveness. At the same time, it also shows that IFEM benefits the representation of prior knowledge.

According to Table 4 we can find:

(1) In the RGBsports data set, compared to the baseball class whose ID number is 0, the detection effect of football is significantly better than that of baseball; that is, AP and R have increased by 0.184 and 0.116, respectively. The possible reason is that the shape and size of the football are easy. Distinguishing from the image background enables the network to learn a more practical difference between the target and the background, thereby improving the detection performance of football. The smaller size of the baseball is easy to confuse with the target background and causes misclassification. In the FLIRs data, the "otherscars" class with ID number 3 achieved the worst performance, probably because this class has a small number of samples, and the slight difference between its "cars" caused misclassification. This reduces the detection accuracy and the performance of the detection framework for the "cars" class. For example, the APs of "dog" and "people" are 0.053 and 0.069higher than "cars," respectively.

(2) In the TTsports dataset, the detection accuracy of each class is very competitive. It is worth noting that although "sportsman6" and "sportsman3" have the same detection performance, AP is 0.838, but R is 0.919 and 0.925, respectively. The possible reason is that "sportsman6" has occlusion during the movement, which reduces the R-value. A visual presentation of the different data is shown in Figure 2.

Reviewer#1, Concern # 4: figures should have high quality. text should be formatted; please add photos of the application of the proposed research, 2-3 photos; what will society have from the paper.

Author response: Thank you very much for the expert’s opinion. I have modified this paper in detail in response to this opinion. At the same time, we modified the paper format.

Author action: We updated the manuscript by.

The main contributions of the LAGSwin detection framework proposed in this paper are as follows:

A local attention-guided Swin-transformer is designed to form mutual embedding between high-level features and low-level features; when the high-level features have insufficient representation ability, many low-level features are embedded in the high-level features. Semantics, when low-level semantic information is weak, embedding a large number of high-level semantics helps resolve the semantic ambiguity between different classes.

A criss-cross fusion strategy is designed to make the target semantics in the low-resolution feature map have a strong representation through cross-fusion. And describe the thermal infrared target from three levels: low, medium, and high, and establish effective spatial relationships and long-term dependency. At the same time, the interaction between different hierarchical features is realized, and the spatial details of sports targets in thermal infrared images are better obtained.

Introducing its convolutional filter in the detection stage, encoding the orientation information while reducing the inconsistency between classification and localization regression and enabling us to generate high-quality anchor and alignment features for Accurate thermal detection of sports objects in infrared images. Finally, the evaluation and demonstration are carried out on the open-source thermal infrared sports dataset and other RGB sports datasets, and the proposed LAGSwin detection framework achieves the best performance in both speed and accuracy. We design a weighted loss function to tune and optimize the proposed framework.

In addition, our proposed thermal infrared moving target detection framework (LAGSwin) can be practically applied to thermal infrared imaging fault diagnosis [24] and other thermal infrared image target detection [25] tasks.

Reviewer#1, Concern # 5: Is there a possibility to use the proposed research for other topics, neural networks, image processing etc.; "Thermographic fault diagnosis of electrical faults of commutator and induction motors"; "Thermographic Fault Diagnosis of Shaft of BLDC Motor”.

Author response: Thank you very much for the expert’s opinion. I have modified this paper in detail in response to this opinion.

Author action: We updated the manuscript by. 

Sports objects may overlap, occlude or shadow during motion. Therefore, capturing more physical appearance attribute information, such as shape and size, in the feature extraction process is necessary to enhance high-level semantics. For example, Masuda T et al. [15] proposed a motion video behavior detection method based on self-supervised feature learning and target detection, which introduced target detection into the process and realized the action detection of multiple people by tracking each person. Considering the high coupling between different features, Zhao J et al. [16] designed a non-global attention mechanism: a local u-shaped attention decoupling network. Jiang X et al. [17] propose a new complementary transformer network (MCNet) for object detection in RGB and thermal infrared images, that is, introduce a transformer-based feature extraction module to efficiently extract hierarchical features of RGB and thermal images and attention-based feature interaction and serial multi-scale dilated convolution (SDC)-based feature fusion module, the complementary interaction of low-level features and semantic fusion of deep features are realized. Liu Z et al. [18] proposed a cross-modal fusion model for GRB and thermal infrared salient target detection - SwinNet. Driven by the Swin Transformer, the method extracts hierarchical features. It bridges the gap between the two modalities driven by the attention mechanism to sharpen salient object contours guided by edge information. Xu F et al. [19][20] considered that due to problems such as color cast and blur in underwater images, the features extracted directly from the backbone network often lack interesting and distinguishable features, which affects the performance of marine target detection. A novel exemplary ocean object detector based on an attention-based spatial pyramid pooling network and bidirectional feature fusion strategy is proposed to alleviate feature weakening and solve the ocean object detection problem. Then, a novel scale-aware feature pyramid structure SA-FPN is proposed to extract rich, robust features of underwater images and improve the performance of marine object detection. Wang H et al. [21][22] aim at minimizing the reconstruction loss between input data and binary codes for autoencoder-based hashing algorithms while ignoring the potential consistency and complementarity of multi-source data, proposes an autoencoder-based multi-view binary clustering hashing algorithm that dynamically learns an associative graph with low-rank constraints, and employs collaborative learning between the autoencoder and the associative graph to learn a unified binary code. Then, considering that most existing methods have to introduce additional clustering steps to produce the final clusters, significantly reducing the unified relationship between graph learning and clustering, a multi-view clustering based on graph collaboration is proposed. Class Methods (MCGC). Xu F et al. [23] considered that the synthetic images are unrealistic enough, affecting the generalization to natural test images. They introduced segmentation masks to construct red, green, and blue mask pairs as input. They also designed an attention-guided style transfer network, learned style features from attention and background regions, and learned content features from entire and attention regions. The feature extraction process considers the target object's lower layers more. Semantics, but the interaction ability between high-level and low-level features is insufficient. At the same time, it is challenging to balance high-level semantics and low-level features when establishing long-term dependencies. Therefore, we propose a local attention-guided Swin-transformer for thermal infrared sports object detection (LAGSwin) to address these limitations. In addition, our proposed thermal infrared moving target detection framework (LAGSwin) can be practically applied to thermal infrared imaging fault diagnosis [24] and other thermal infrared image target detection [25] tasks.

[24] Glowacz A. Thermographic fault diagnosis of electrical faults of commutator and induction motors[J]. Engineering Applications of Artificial Intelligence, 2023, 121: 105962.

[25] Glowacz A. Thermographic fault diagnosis of shaft of BLDC motor[J]. Sensors, 2022, 22(21): 8537.

Reviewer#1, Concern # 6: references should be from the web of science 2020-2023 (50% of all references, 30 references at least); please compare advantages/disadvantages of other approaches.

Author response: Thanks a lot for the expert opinion. In response to this opinion, I made detailed revisions to this paper. In addition, we expanded the references to 37.

Author action: We updated the manuscript by. 

With the successful application of deep learning technology in many fields, object detection has developed significantly in recent years. At present, the standard thermal infrared target detection algorithms are mainly divided into two types, namely, two-stage detection and single-stage detection methods. Among them, the two-stage detection method realizes feature extraction by generating a sparse ROI set and performs boundary regression and object classification in the second stage. For example, Li et al. [8] designed a faster light-sensing two-stage RCNN detection model for the differences between optical and thermal infrared images. They discussed the feature extraction capabilities of various convolutional networks in depth. Aiming at the problem of reliable and efficient object detection in thermal infrared images, Dai X et al. [5] propose a novel object detection method based on convolutional networks, which can be optimized and predicted in an end-to-end manner. Dai et al. [9] proposed a multi-task Faster RCNN detector to evaluate the driving distance to improve driving safety. They improved the performance of thermal infrared object detection tasks by adjusting the feature extractor. Song et al. [4] created a segmentation template for the heat-generating part for the heat-generating components in the thermal-sensing image of a thermal infrared camera. They proposed a mask-based RCNN-based infrared image detection algorithm. Although the detection progress of these two-stage thermal infrared target detection algorithms is good, the model efficiency needs to be improved. At the same time, these methods often use a relatively simple convolution structure in the feature extraction stage and pay too much attention to thermal infrared images in the feature capture stage. The high-level semantics of the target ignores rich low-level semantic information.

Xue Y et al. [14] used compressed Darknet53 to obtain the feature information of two modalities. They used a weighted fusion strategy for feature fusion, proposing a thermal infrared pedestrian detection algorithm with multi-modal attention fusion. Although these thermal infrared detection methods have good detection efficiency, the detection accuracy needs to be improved. At the same time, they mainly use traditional convolution methods or simple weighted fusion strategies in the feature extraction stage, which are often quickly introduced in the feature transfer process—a large amount of redundant information. In addition, these methods mainly focus on detecting single structural targets in thermal infrared images and less on multi-category sports targets.

Wang H et al. [21][22] aim at minimizing the reconstruction loss between input data and binary codes for autoencoder-based hashing algorithms while ignoring the potential consistency and complementarity of multi-source data, proposes an autoencoder-based multi-view binary clustering hashing algorithm that dynamically learns an associative graph with low-rank constraints, and employs collaborative learning between the autoencoder and the associative graph to learn a unified binary code. Then, considering that most existing methods have to introduce additional clustering steps to produce the final clusters, significantly reducing the unified relationship between graph learning and clustering, a multi-view clustering based on graph collaboration is proposed. Class Methods (MCGC). Xu F et al. [23] considered that the synthetic images are unrealistic enough, affecting the generalization to natural test images. They introduced segmentation masks to construct red, green, and blue mask pairs as input. They also designed an attention-guided style transfer network, learned style features from attention and background regions, and learned content features from entire and attention regions. The feature extraction process considers the target object's lower layers more. Semantics, but the interaction ability between high-level and low-level features is insufficient. At the same time, it is challenging to balance high-level semantics and low-level features when establishing long-term dependencies. Therefore, we propose a local attention-guided Swin-transformer for thermal infrared sports object detection (LAGSwin) to address these limitations.

Reviewer#1, Concern # 7: Conclusion: point out what have you done; please add some sentences about future work.

Author response: Thank you very much for the expert’s opinion. I have modified this paper in detail in response to this opinion.

Author action: We updated the manuscript by. 

In this paper, we propose a local attention-guided swain-transformer detection framework (LAGSwin) for detecting spatial details of moving objects in thermal infrared images. The method first uses a feature extraction module guided by local attention to strengthen the interaction between low-level and high-level features. It embeds high-level features into low-level features so that high-level features contain rich low-level semantics. Embedding high-level features into low-level features makes low-level features more discriminative in high-level semantics. Secondly, design a cross-fusion strategy to aggregate this feature information from different directions, reduce redundant information while retaining spatial details to the greatest extent, and ensure the integrity and diversity of attribute information such as the physical appearance of the target; in the detection module, complete The feature alignment of the algorithm alleviates the inconsistency between regression and classification. Finally, evaluation tests were performed on three sets of open-source baseline data, including TTsports, FLIRs, and RGBsports, and optimal detection performance and good robustness were achieved.

During the experiment, we found that the design of the feature extraction module of the detection framework is relatively complicated, which increases the redundancy of the model. At the same time, the detection efficiency also has a lot of room for improvement. Therefore, in the following research, we will start from the above two aspects to design a simple and efficient semantic guidance network, that is, to lighten the feature extractor and design a more effective semantic fusion module. , to preserve the spatial details of the target in the thermal infrared image to the greatest extent, and at the same time, use the new semantic fusion module to efficiently gather different levels of semantics to improve feature representation performance.

Reviewer#2. This manuscript proposed a local attention-guided Swin-transformer thermal infrared sports object detection method (LAGSwin) to encode sports objects’ spatial transformation and orientation information. The evaluation and verification of multiple open-source sports target datasets prove that the proposed LAGSwin detection framework has good robustness and generalization ability. However, there are some concerns about this manuscript:

Concern # 1: In related work, the author introduced the one-stage detector and the two-stage detector. It is recommended that the transformer detector be further supplemented.

Author response: Thank you very much for the expert’s opinion. I have modified this paper in detail in response to this opinion.

Author action: We updated the manuscript by.

Jiang X et al. [17] propose a new complementary transformer network (MCNet) for object detection in RGB and thermal infrared images, that is, introduce a transformer-based feature extraction module to efficiently extract hierarchical features of RGB and thermal images and attention-based feature interaction and serial multi-scale dilated convolution (SDC)-based feature fusion module, the complementary interaction of low-level features and semantic fusion of deep features are realized. Liu Z et al. [18] proposed a cross-modal fusion model for GRB and thermal infrared salient target detection - SwinNet. Driven by the Swin Transformer, the method extracts hierarchical features. It bridges the gap between the two modalities driven by the attention mechanism to sharpen salient object contours guided by edge information.

[17] Jiang X, Zhu L, Hou Y, et al. Mirror complementary transformer network for RGB-thermal salient object detection[J]. arXiv preprint arXiv:2207.03558, 2022.

[18] Liu Z, Tan Y, He Q, et al. SwinNet: Swin transformer drives edge-aware RGB-D and RGB-T salient object detection[J]. IEEE Transactions on Circuits and Systems for Video Technology, 2021, 32(7): 4486-4497.

Reviewer#2, Concern # 2: In ablation experiment, the ablation results of local attention guided feature extraction module, criss-cross fusion module and detection module for aligned convolutional should be provided to justify the detailed design.

Author response: Thank you so much for your expert opinion. Table 2 presents the experimental results of different modules.

Author action: We updated the manuscript by. 

From Table 2, we draw the following conclusions:

(3) Using Res2net101 as the backbone network to replace the Swin-transformer in our model has achieved the best competitiveness. At the same time, on the RGBsports data, the mAP is 0.004 higher than that of the Swin-transformer. This may be because, in the local feature capture stage, res2net101 first divides the target features in the RGB image into multiple subspaces so that the network can obtain more detailed local semantics. Still, the R value is low, which may be the model with the deepening of the number of network layers leads to the utilization of a large amount of redundant information, which weakens the representation of the global semantics of the target. The model may also have fallen into a local optimum, and overfitting has occurred.

(4) IFEM+Backbone+CCFM+ACDM showed the worst performance on TTsports, RGBsports, and FLIRs data sets. The possible reason is significant spatial distribution and physical meaning differences between different scales. If you directly use the simple fusion method, It is challenging to balance the differences between them. Still, the LAG layer we designed plays an essential role using the weight distribution and residual strategy. IFEM+Backbone+LAG+PAFPN+ACDM has gained a better competitive advantage than IFEM+Backbone+LAG+FPN+ACDM. This may be because PAFPN also uses its own. The top-down and bottom-up two-way information transmission strategy reduces the loss of details caused by information transmission while preserving the global semantics to the greatest extent. In addition, the IFEM+Backbone+LAG+ACDM method achieved the worst performance on the three sets of data sets, with mAPs of 0.808, 0.802, and 0.805, respectively, which shows that CCFM plays a positive role in the overall framework and is beneficial to the contextual detail features. The Backbone+LAG+CCFM+ACDM method has achieved better competitiveness. At the same time, it also shows that IFEM benefits the representation of prior knowledge.

Reviewer#2, Concern # 3: Subjective experiment content is insufficient, which needs to be further supplemented to improve the credibility of the work.

Author response: Thanks a lot for the expert opinion. Table 1 presents the experimental results of different state-of-the-art methods, and table 2 presents the experimental results of different modules..

Author action: We updated the manuscript by.

From Table 1, we can draw the following conclusions:

(2) Compared with single-stage detection methods such as SASM, Oriented RepPoints, and KLD, two-stage detection methods such as ReDet, Roi-transformer, and Oriented R-CNN have achieved strong competitiveness on these open source datasets. For example, the mAP of the Roi-transformer is 0.079, 0.039, and 0.039 higher than the Oriented RepPoints detection method, respectively. The mAP of Oriented R-CNN is 0.06, 0.068, and 0.055, higher than that of the KLD detection method. It may be that the two-stage detection method generates high-quality regions of interest in the first stage, which prompts the network to learn a better feature representation. In addition, the SASM detection method performed the worst on the three open-source datasets. The possible reason is that the SASM method focuses on the representation of object shape information while ignoring the extraction of high-level discriminative semantics. It is worth noting that the single-stage detection method Oriented RepPoints has achieved better competitiveness on the RGBsports and FLIRs datasets. For example, the mAP of Oriented RepPoints is 0.026 and 0.036 higher than Oriented R-CNN, respectively. It is possible that the Oriented RepPoints detection method uses adaptive point representation and dynamic evaluation and allocation strategies, which promotes the network to capture any instance-oriented geometric information effectively, and uses three orientation conversion functions to achieve accurate positioning of the target, and at the same time filters the feature point set Highlighting the representation improves the classification accuracy, which finally leads to the Oriented RepPoints detection method outperforming the Oriented R-CNN method. In addition, the current advanced single-stage detection algorithm Rtmdet and two-stage detection algorithm DiffusionDet have achieved good competitive advantages in the three datasets, such as mAP on the FLIRs data set are 0.815 and 0.799, respectively. The detection efficiency of Rtmdet is better than DiffusionDet.

(3) The proposed LAGSwin detection framework is still highly competitive in reasoning efficiency while ensuring optimal detection accuracy. For example, the FLOPs of LAGSwin are 1.8, 2.4, and 1.3 lower than single-stage detection methods such as Oriented RepPoints, SASM, and KLD, respectively. Still, our detection accuracy is far superior to these methods. Compared with the two-stage detection method, our proposed LAGSwin detection framework achieves the best detection accuracy and inference efficiency (FLOPs).

From Table 2, we draw the following conclusions:

(1) In our proposed LAGSwin thermal infrared sports object detection framework, each component plays a crucial positive role in the overall performance of the framework. On the three open-source data sets, the backbone network using PVTV2 as the model has achieved a tremendous competitive advantage, such as the mAP of Resnet101+DCN increased by 0.01, 0.01, and 0.001, respectively. On the TTsports and FLIRs data sets, compared with Res2net101 mAP increased by 0.002 and 0.004, respectively, but decreased by 0.006 on the RGBsports dataset. The main reason is that PVTV2 benefits from the self-attention mechanism in Transformer and always maintains the global receptive field, ensuring the local semantics of the target and better acquisition of the target. The global details of the RGBsports data set may be reduced because the target scale changes significantly or the target background is more complex, which reduces the detection performance of PVTV2. In addition, HRNet obtained the worst detection performance, which was 0.007, 0.009, and 0.002 lower than MobileNetV2 on the three open-source datasets. The possible reason is that HRNet focuses on obtaining high-resolution feature information of the target, ignoring the rich low-resolution feature information. Hierarchical semantic information also shows that low-level semantic information helps detect thermal infrared sports targets.

(4) IFEM+Backbone+CCFM+ACDM showed the worst performance on TTsports, RGBsports, and FLIRs data sets. The possible reason is significant spatial distribution and physical meaning differences between different scales. If you directly use the simple fusion method, It is challenging to balance the differences between them. Still, the LAG layer we designed plays an essential role using the weight distribution and residual strategy. IFEM+Backbone+LAG+PAFPN+ACDM has gained a better competitive advantage than IFEM+Backbone+LAG+FPN+ACDM. This may be because PAFPN also uses its own. The top-down and bottom-up two-way information transmission strategy reduces the loss of details caused by information transmission while preserving the global semantics to the greatest extent. In addition, the IFEM+Backbone+LAG+ACDM method achieved the worst performance on the three sets of data sets, with mAP of 0.808, 0.802, and 0.805, respectively, which shows that CCFM plays a positive role in the overall framework and is beneficial to the contextual detail features. The Backbone+LAG+CCFM+ACDM method has achieved better competitiveness. At the same time, it also shows that IFEM benefits the representation of prior knowledge.

Reviewer#2, Concern # 4: In comparison experiments, it is recommended to provide fair comparison results with the state-of-the-art schemes.

Author response: Thanks a lot for the expert opinion. Table 1 presents the experimental results of different state-of-the-art methods.

Author action: We updated the manuscript by. 

To ensure the smooth progress of the experiments, all experiments were completed on 4 RTX3090 of python3.7.6 and torch1.7.0+cu110, and the recall rate and mean average precision rate (mAP) were used as evaluation indicators. The calculation process is shown in the equation.

From Table 1, we can draw the following conclusions:

(1) The thermal infrared sports target detection framework of LAGSwin we proposed has achieved the best detection performance on three open-source datasets, including TTsports, RGBsports, and FLIRs. For example, the mAP on the TTsports, RGBsports, and FLIRs data sets is 0.057, 0.017, and 0.025 higher than the ReDet method. The possible reason is that, on the one hand, we describe the targets in thermal infrared images in detail from three different scales and levels of low, medium, and high, and establish functional spatial and long-term dependencies between these features and use The local attention guidance layer highlights the details, allowing the network to better focus on the subtle changes between different types of objects, as well as the differences between objects and backgrounds. On the other hand, introducing convolutional filters in the detector, encoding orientation information while reducing the inconsistency between classification and localization regression, enables us to generate high-quality anchor and alignment features. In addition, each component assists the network in obtaining the optimal feature representation, which ultimately leads to the optimal performance of the proposed model on the three datasets.

(2) Compared with single-stage detection methods such as SASM, Oriented RepPoints, and KLD, two-stage detection methods such as ReDet, Roi-transformer, and Oriented R-CNN have achieved strong competitiveness on these open source datasets. For example, the mAP of the Roi-transformer is 0.079, 0.039, and 0.039 higher than the Oriented RepPoints detection method, respectively. The mAP of Oriented R-CNN is 0.06, 0.068, and 0.055, higher than that of the KLD detection method. It may be that the two-stage detection method generates high-quality regions of interest in the first stage, which prompts the network to learn a better feature representation. In addition, the SASM detection method performed the worst on the three open-source datasets. The possible reason is that the SASM method focuses on the representation of object shape information while ignoring the extraction of high-level discriminative semantics. It is worth noting that the single-stage detection method Oriented RepPoints has achieved better competitiveness on the RGBsports and FLIRs datasets. For example, the mAP of Oriented RepPoints is 0.026 and 0.036 higher than Oriented R-CNN, respectively. It is possible that the Oriented RepPoints detection method uses adaptive point representation and dynamic evaluation and allocation strategies, which promotes the network to capture any instance-oriented geometric information effectively, and uses three orientation conversion functions to achieve accurate positioning of the target, and at the same time filters the feature point set Highlighting the representation improves the classification accuracy, which finally leads to the Oriented RepPoints detection method outperforming the Oriented R-CNN method. In addition, the current advanced single-stage detection algorithm Rtmdet and two-stage detection algorithm DiffusionDet have achieved good competitive advantages in the three datasets, such as mAP on the FLIRs data set are 0.815 and 0.799, respectively. The detection efficiency of Rtmdet is better than DiffusionDet.

(3) The proposed LAGSwin detection framework is still highly competitive in reasoning efficiency while ensuring optimal detection accuracy. For example, the FLOPs of LAGSwin are 1.8, 2.4, and 1.3 lower than single-stage detection methods such as Oriented RepPoints, SASM, and KLD, respectively. Still, our detection accuracy is far superior to these methods. Compared with the two-stage detection method, our proposed LAGSwin detection framework achieves the best detection accuracy and inference efficiency (FLOPs).

Reviewer#2, Concern # 5: For the task of object detection and attention, some recent works should be discussed, including "Refined marine object detector with attention-based spatial pyramid pooling networks and bidirectional feature fusion strategy", "Scale-aware feature pyramid architecture for marine object detection", "Graph-Collaborated Auto-Encoder Hashing for Multi-view Binary Clustering" and "Towards Adaptive Consensus Graph: Multi-view Clustering via Graph Collaboration", "Purifying real images with an attention-guided style transfer network for gaze estimation".

Author response: Thank you very much for the expert’s opinion. I have modified this paper in detail in response to this opinion.

Author action: We updated the manuscript by.

Xu F et al. [19][20] considered that due to problems such as color cast and blur in underwater images, the features extracted directly from the backbone network often lack interesting and distinguishable features, which affects the performance of marine target detection. A novel exemplary ocean object detector based on an attention-based spatial pyramid pooling network and bidirectional feature fusion strategy is proposed to alleviate feature weakening and solve the ocean object detection problem. Then, a novel scale-aware feature pyramid structure SA-FPN is proposed to extract rich, robust features of underwater images and improve the performance of marine object detection. Wang H et al. [21][22] aim at minimizing the reconstruction loss between input data and binary codes for autoencoder-based hashing algorithms while ignoring the potential consistency and complementarity of multi-source data, proposes an autoencoder-based multi-view binary clustering hashing algorithm that dynamically learns an associative graph with low-rank constraints, and employs collaborative learning between the autoencoder and the associative graph to learn a unified binary code. Then, considering that most existing methods have to introduce additional clustering steps to produce the final clusters, significantly reducing the unified relationship between graph learning and clustering, a multi-view clustering based on graph collaboration is proposed. Class Methods (MCGC). Xu F et al. [23] considered that the synthetic images are unrealistic enough, affecting the generalization to natural test images. They introduced segmentation masks to construct red, green, and blue mask pairs as input. They also designed an attention-guided style transfer network, learned style features from attention and background regions, and learned content features from entire and attention regions.

Reviewer#2, Concern # 6: We would like to know the comparison results (accuracy, parameters, computations) of the proposed algorithm with the current popular one-stage, two-stage and transformer detectors.

Author response: Thanks a lot for the expert opinion. Table 1 presents the experimental results of different state-of-the-art methods.

Author action: We updated the manuscript by.

(2) Compared with single-stage detection methods such as SASM, Oriented RepPoints, and KLD, two-stage detection methods such as ReDet, Roi-transformer, and Oriented R-CNN have achieved strong competitiveness on these open source datasets. For example, the mAP of the Roi-transformer is 0.079, 0.039, and 0.039 higher than the Oriented RepPoints detection method, respectively. The mAP of Oriented R-CNN is 0.06, 0.068, and 0.055, higher than that of the KLD detection method. It may be that the two-stage detection method generates high-quality regions of interest in the first stage, which prompts the network to learn a better feature representation. In addition, the SASM detection method performed the worst on the three open-source datasets. The possible reason is that the SASM method focuses on the representation of object shape information while ignoring the extraction of high-level discriminative semantics. It is worth noting that the single-stage detection method Oriented RepPoints has achieved better competitiveness on the RGBsports and FLIRs datasets. For example, the mAP of Oriented RepPoints is 0.026 and 0.036 higher than Oriented R-CNN, respectively. It is possible that the Oriented RepPoints detection method uses adaptive point representation and dynamic evaluation and allocation strategies, which promotes the network to capture any instance-oriented geometric information effectively, and uses three orientation conversion functions to achieve accurate positioning of the target, and at the same time filters the feature point set Highlighting the representation improves the classification accuracy, which finally leads to the Oriented RepPoints detection method outperforming the Oriented R-CNN method. In addition, the current advanced single-stage detection algorithm Rtmdet and two-stage detection algorithm DiffusionDet have achieved good competitive advantages in the three datasets, such as mAP on the FLIRs data set are 0.815 and 0.799, respectively. The detection efficiency of Rtmdet is better than DiffusionDet.

(3) The proposed LAGSwin detection framework is still highly competitive in reasoning efficiency while ensuring optimal detection accuracy. For example, the FLOPs of LAGSwin are 1.8, 2.4, and 1.3 lower than single-stage detection methods such as Oriented RepPoints, SASM, and KLD, respectively. Still, our detection accuracy is far superior to these methods. Compared with the two-stage detection method, our proposed LAGSwin detection framework achieves the best detection accuracy and inference efficiency (FLOPs).

---

## [Decision Letter · Decision Letter 1]

22 Nov 2023

PONE-D-23-13636R1LAGSwin: Local attention guided Swin-transformer for thermal infrared sports object detectionPLOS ONE

Dear Dr. Mao,

Thank you for submitting your manuscript to PLOS ONE. After careful consideration, we feel that it has merit but does not fully meet PLOS ONE’s publication criteria as it currently stands. Therefore, we invite you to submit a revised version of the manuscript that addresses the points raised during the review process.

We look forward to receiving your revised manuscript.

Kind regards,

Nattapol Aunsri, Ph.D.

Academic Editor

PLOS ONE

Journal Requirements:

**Additional Editor Comments:**

Please consider the comment from reviewer 3 carefully before submitting your revision.

Reviewers' comments:

Reviewer's Responses to Questions

**Comments to the Author**

1. If the authors have adequately addressed your comments raised in a previous round of review and you feel that this manuscript is now acceptable for publication, you may indicate that here to bypass the “Comments to the Author” section, enter your conflict of interest statement in the “Confidential to Editor” section, and submit your "Accept" recommendation.

Reviewer #1: All comments have been addressed

Reviewer #3: (No Response)

2. Is the manuscript technically sound, and do the data support the conclusions?

Reviewer #1: Yes

Reviewer #3: Partly

3. Has the statistical analysis been performed appropriately and rigorously? 

Reviewer #1: Yes

Reviewer #3: N/A

4. Have the authors made all data underlying the findings in their manuscript fully available?

Reviewer #1: Yes

Reviewer #3: Yes

5. Is the manuscript presented in an intelligible fashion and written in standard English?

Reviewer #1: Yes

Reviewer #3: Yes

6. Review Comments to the Author

Reviewer #1: The paper is good enough to publish.

Reviewer #3: 1. In the discussion section, there is a reference to Figure 2 in the discussion of the results. However, Figure 2 was not located inside the document itself (revision1 version).

"line 472 - framework, Figure 2 and Table 4 show the detection performance of the model for each"

Please examine the document for clarity and completeness by ensuring that all related figures are included.

This visual assistance would greatly improve understanding of your experimental outcomes as well as the efficiency of the LAGSwin method across various sports targets.

2. Please use proper formatting throughout the manuscript, particularly with items such as double quotation marks (" "). Check the consistency and accuracy of these symbols.

7. PLOS authors have the option to publish the peer review history of their article (what does this mean?). If published, this will include your full peer review and any attached files.

Reviewer #1: No

Reviewer #3: No

---

## [Author Response · Author response to Decision Letter 1]

7 Dec 2023

1. In the discussion section, there is a reference to Figure 2 in the discussion of the results. However, Figure 2 was not located inside the document itself (revision1 version). "line 472 - framework, Figure 2 and Table 4 show the detection performance of the model for each". Please examine the document for clarity and completeness by ensuring that all related figures are included. This visual assistance would greatly improve understanding of your experimental outcomes as well as the efficiency of the LAGSwin method across various sports targets.

Answer. We are very grateful to the experts for their constructive comments. We have made detailed revisions in response to these comments. 

For instance, 

(2) In the TTsports dataset, the detection accuracy of each class is very competitive. It is worth noting that although and have the same detection performance, AP is , but R is and , respectively. The possible reason is that has occlusion during the movement, which reduces the R-value. A visual presentation of the different data is shown in Figure 2.

2. Please use proper formatting throughout the manuscript, particularly with items such as double quotation marks (" "). Check the consistency and accuracy of these symbols.

Answer. We are very grateful to the experts for their constructive comments. We have made detailed revisions in response to these comments. 

For instance, 

(1) In the RGBsports data set, compared to the baseball class whose ID number is 0, the detection effect of football is significantly better than that of baseball, namely, AP and R have increased by and , respectively. The possible reason is that the shape and size of the football are easy. Distinguishing from the image background enables the network to learn a more practical difference between the target and the background, thereby improving the detection performance of football. The smaller size of the baseball is easy to confuse with the target background and causes misclassification. In the FLIRs data, the class with ID number 3 achieved the worst performance, probably because this class has a small number of samples, and the slight difference between its caused misclassification. This reduces the detection accuracy and the performance of the detection framework for the class. For example, the APs of and are and higher than , respectively.

---

## [Decision Letter · Decision Letter 2]

28 Dec 2023

LAGSwin: Local attention guided Swin-transformer for thermal infrared sports object detection

PONE-D-23-13636R2

Dear Dr. Mao,

We’re pleased to inform you that your manuscript has been judged scientifically suitable for publication and will be formally accepted for publication once it meets all outstanding technical requirements.

Kind regards,

Nattapol Aunsri, Ph.D.

Academic Editor

PLOS ONE

Additional Editor Comments (optional):

Reviewers' comments:

Reviewer's Responses to Questions

**Comments to the Author**

1. If the authors have adequately addressed your comments raised in a previous round of review and you feel that this manuscript is now acceptable for publication, you may indicate that here to bypass the “Comments to the Author” section, enter your conflict of interest statement in the “Confidential to Editor” section, and submit your "Accept" recommendation.

Reviewer #3: All comments have been addressed

2. Is the manuscript technically sound, and do the data support the conclusions?

Reviewer #3: Yes

3. Has the statistical analysis been performed appropriately and rigorously? 

Reviewer #3: Yes

4. Have the authors made all data underlying the findings in their manuscript fully available?

Reviewer #3: Yes

5. Is the manuscript presented in an intelligible fashion and written in standard English?

Reviewer #3: Yes

6. Review Comments to the Author

Reviewer #3: I would recommend to accept the manuscript as the authors have addressed all the suggested recommendations to improve the manuscript as per journal guidelines and quality.

7. PLOS authors have the option to publish the peer review history of their article (what does this mean?). If published, this will include your full peer review and any attached files.

Reviewer #3: No

---

## [Editor Report · Acceptance letter]

23 Jan 2024

PONE-D-23-13636R2 

PLOS ONE

Dear Dr. Mao, 

I'm pleased to inform you that your manuscript has been deemed suitable for publication in PLOS ONE. Congratulations! Your manuscript is now being handed over to our production team.

Kind regards, 

on behalf of

Dr. Nattapol Aunsri 

Academic Editor

PLOS ONE